# Autofluorescence and Metabotyping of Soybean Varieties Using Confocal Laser Microscopy and High-Resolution Mass Spectrometric Approaches

**DOI:** 10.3390/plants14131995

**Published:** 2025-06-30

**Authors:** Mayya P. Razgonova, Muhammad A. Navaz, Ekaterina S. Butovets, Ludmila M. Lukyanchuk, Olga A. Chunikhina, Sezai Ercişli, Alexei N. Emelyanov, Kirill S. Golokhvast

**Affiliations:** 1N.I. Vavilov All-Russian Institute of Plant Genetic Resources, B. Morskaya 42-44, 190000 Saint-Petersburg, Russia; chunikhina_o@mail.ru (O.A.C.); golokhvast@sfsca.ru (K.S.G.); 2Advanced Engineering School “Institute of Biotechnology, Bioengineering and Food Systems”, Far Eastern Federal University (FEFU), Fr. Russian, pos. Ajax, 10, 690922 Vladivostok, Russia; 3Laboratory for Research and Application of Supercritical Fluid Technologies in Agro-Food Biotechnology, National Research Tomsk State University, Lenin Ave, 36, 634050 Tomsk, Russia; 4Advanced Engineering School «Agrobiotek», National Research Tomsk State University, Lenin Ave, 36, 634050 Tomsk, Russia; 5Federal Scientific Center of Agricultural Biotechnologies of the Far East Named After A.K. Chaika, St. Volozhenina, 30, Timiryazevsky Village, 692539 Ussuriysk, Russia; otdelsoy@mail.ru (E.S.B.); lucy1987@mail.ru (L.M.L.); emelyanov.prim@yandex.ru (A.N.E.); 6Department of Horticulture, Faculty of Agriculture, Ataturk University, Erzurum 25240, Türkiye; sercisli@atauni.edu.tr; 7Siberian Federal Scientific Centre of Agrobiotechnology, RAS, Centralnaya, Presidium, 633501 Krasnoobsk, Russia

**Keywords:** *Glycine max* (L.) Merr., soybean, laser microscopy, tandem mass spectrometry, polyphenolic compounds

## Abstract

This research examines a detailed metabolomic and comparative analysis of bioactive substances of soybean varieties: “Primorskaya-4”, “Primorskaya-86”, “Primorskaya-96”, “Locus”, “Sphere”, “Breeze”, “Namul”, and “Musson” by the laser confocal microscope CLSM 800 and the mass spectrometry of bioactive compounds by tandem mass spectrometry. The laser microscopy allowed us to clarify in detail the spatial arrangement of phenolic acids, flavonols, and anthocyanin contents in soybeans. Research has convincingly shown that the polyphenolic content of soybeans, and, in particular, the anthocyanins, are spatially localized mainly in the seed coat of soybeans. Tandem mass spectrometry was used to identify chemical constituents in soybean extracts. The results of initial studies revealed the presence of one hundred and fourteen compounds; sixty-nine of the target analytes were tentatively identified as compounds from polyphenol groups.

## 1. Introduction

Currently, the nutritional qualities of agricultural crops have received much more attention in terms of the quality of life and health of potential consumers [1]. Antioxidant activity has been widely discussed regarding the nutritional value of various crops as it plays a crucial role in the prevention of several chronic diseases [2]. Antioxidant activity is largely determined by the type and content of different compounds of the polyphenol group, such as anthocyanins, tannins, flavonoids, etc. [3,4]. The study of phytochemical antioxidants can help improve the nutritional properties of crops to meet human health needs. Long-term crop breeding for high-yielding traits has significantly reduced the diversity of genes associated with nutritional quality [5]. Soybeans, a staple food worldwide, are not only a valuable source of oil and protein but are also rich in health-promoting polyphenolic compounds and soya saponins [6]. Isoflavonoids and soy saponins are well-known phytochemicals in soybeans with a wide range of biological activities against oxidative stress-related disorders [7]. Taken together, soybean is a good model crop to study the diversity of functional antioxidants and identify genes associated with chemical synthesis and decoration. During soybean domestication, one of the most obvious changes is the difference in seed coat pigmentation [8]. Genes and transcription factors associated with anthocyanin biosynthesis contributed to the formation of pigmented seed coats in soybean [9]. These data suggest that pigmented wild soybeans have a high metabolic diversity of polyphenolic anthocyanin precursors and products. Anthocyanins are a well-studied class of flavonoids [10]. Flavonoids are a large class of polyphenols and can be subdivided into flavones, flavonols, flavanones, flavanols, chalcones, aurones, isoflavones, anthocyanidins, etc. [11]. The multiple phenolic hydroxyl groups in the backbone of flavonoids contribute to their potent antioxidant activity [12]. In addition to the well-studied isoflavones and anthocyanidins, the emerging characterization of the effects of domestication of other flavonoid subclasses can facilitate the use of more polyphenolic antioxidants in both the food and dietary supplement industries. Natural chemical modifications such as glycosylation and acylation alter the polarity, solubility, stability, bioavailability, and biological activity of polyphenolic antioxidants [13,14]. Due to the significant impact of modifications on the functional properties of polyphenolic antioxidants, it is important to characterize modifications of polyphenols to achieve functional improvement in various soy food products.

It should also be noted that soybean *Glycine max* (L.) Merr. and soybean-based foods are the main natural sources of saponins in dietary and functional nutrition [15,16]. Soy saponins belong to the group of triterpene glycosides and are divided into three main groups based on the differences in the substitution of the C-22 and C-23 positions of the aglycone (or soyasapogenol): group A, B, and E soy saponins. Group A soy saponins have a glycosyl chain attached to the C-3 and C-22 positions of the aglycone. Group B soy saponins have only one glycosyl chain (attached to the C-3 position) and can be conjugated with 2,3-dihydro-2,5-dihydroxy-6-methyl-4H-pyran-4-one (DDMP) at the C-22 position [17]. The E group of soybean saponins is the least abundant and is considered to be photooxidation products of the B group of soybean saponins [18]. In soybeans, the A group of soybean saponins is considered to be fully acetylated [18,19]. Soybean saponins are important bioactive components with significant beneficial effects on human health. The biological effects of soybean saponins have been widely described and include hepatoprotective, antitumor, immunostimulatory, antiviral, and hypocholesterolemic activities [20,21]. The soybean saponin group is considered to be responsible for the astringent and bitter taste of soybean foods, mainly due to the presence of acetyl groups [22]. The exact mechanisms underlying the biological properties of soybean saponins remain to be elucidated due to the lack of purified test compounds and limited information on the content and composition of soybean saponins in soybeans and soybean products. The quantitative determination of individual soybean saponins has always been a difficult task, partly due to the difficulties in isolating authentic standards and the structural complexity of this group of phytochemicals. The covalent bonds linking the acetyl groups, and especially the DDMP groups, to the saponin molecule are relatively weak, even under relatively mild extraction conditions, making it difficult to obtain saponins in their native form [23].

In addition, new advanced research methods are becoming more widespread, such as laser microscopy, a method that exploits the ability of chemicals to fluoresce when excited by a laser. This can be used to solve various visualization problems. Previous autofluorescence-based microscopic studies of soybean plants have focused more on the visualization of anatomical features: the three-dimensional (3D) internal structure of a soybean seed [24] and leaf anatomy of *Glycine max* (L.) Merr. [25]. We propose the autofluorescence-based study of the spatial distribution of some groups of phytochemicals in the seed tissue using confocal laser microscopy.

This study represents a comparative analysis of eight soybean varieties cultivated in the N.I. Vavilov All-Russian Institute of Plant Genetic Resources and the A.K. Chaika Federal Scientific Centre of Agrobiotechnologies of the Far East. A detailed metabolomic analysis was carried out using tandem mass spectrometry.

Detailed metabolomic and comparative analyses of bioactive substances of soybean varieties were carried out: “Primorskaya-4”, “Primorskaya-86”, “Primorskaya-96”, “Locus”, “Sphere”, “Breeze”, “Namul”, and “Musson”, cultivated in the Federal Scientific Centre of Agrobiotechnologies of the Far East named by A.K. Chaika and in N.I. Vavilov All-Russian Institute of Plant Genetic Resources by means of the laser confocal microscope CLSM 800 and the mass spectrometry of bioactive compounds by ion trap amaZon SL.

## 2. Materials and Methods

### 2.1. Plant Material

Eight soybean varieties were evaluated. The varieties (“Primorskaya 4”, “Primorskaya 86”, “Primorskaya 96”, “Locus”, “Sphere”, “Breeze”, “Namul”, “Musson”) were collected and grown in the Far East Federal Scientific Centre of Agrobiotechnologies named after A.K. Chaika, following standard agronomic practices. The soybeans were harvested at the end of September 2022. Triplicate (250 g each) samples were taken for each variety. Only completely healthy seed samples were considered for further analysis. Samples were washed with distilled water, dried at room temperature, and stored at −80 °C until processing. All samples conformed morphologically to the pharmacopoeial standards of the State Pharmacopoeia of the Russian Federation [26].

### 2.2. Chemicals and Reagents and Fractional Maceration

Analytical grade reagents and ultrapure water were used for liquid chromatography (LC) and mass spectrometry (MS).

Highly concentrated extracts were prepared using fractional maceration, as reported earlier [27]. For each replicate of the varieties, 50 g of seeds were macerated and extracted with ethanol (95%). Infusions were prepared as per the methods described in our earlier report [27]. Moreover, the extraction was carried out in triplicate, followed by filtering through a Whatman filter paper. Finally, we used acetonitrile for preparing the final working concentration for LC and MS analyses.

### 2.3. Liquid Chromatography and Mass Spectrometry

Mass spectrometry analysis was performed on an ion trap amaZon SL (BRUKER DALTONIKS, Bremen, Germany) equipped with an ESI source in positive or/and negative ion modes. The optimized parameters were obtained as follows: ionization source temperature: 70 °C, gas flow: 4 L/min, nebulizer gas (atomizer): 7.3 psi, capillary voltage: 4500 V, end plate bend voltage: 1500 V, fragmentary: 280 V, collision energy: 60 eV. An ion trap was used in the scan range of *m*/*z* 100–1.700 for MS and MS/MS. The chemical constituents were identified by comparing their retention index, mass spectra, and MS fragmentation with an in-house self-built database (Biotechnology, Bioengineering and Food Systems Laboratory, Far-Eastern Federal University, Vladivostok, Russia). The in-house self-built database was based on data from other spectroscopic techniques, such as nuclear magnetic resonance, ultraviolet spectroscopy, and MS, as well as data from the literature, which is continuously updated and revised. The capture rate was one spectrum for MS and two spectra for MS/MS. Data acquisition was controlled by Windows software for BRUKER DALTONIKS. All experiments were repeated three times. A four-stage ion separation mode (MS/MS mode) was implemented.

### 2.4. Optical Microscopy

Optical microscopy was carried out according to the method described earlier [27]. Briefly, soybean seeds were dissected, their autofluorescence parameters were determined using the laser confocal microscope CLSM 800 (Zeiss, Germany), and fluorescence maxima were registered by excitation with violet and blue lasers at respective emission ranges. After the excitation, the images were taken using ZEN 2.1 software (Carl Zeiss Microscopy GmbH, Jena, Germany).

### 2.5. Statistical Analysis

The upset plot was prepared using an online tool ChiPlot (https://www.chiplot.online/; accessed on 1 December 2024). Principal component analysis, based on covariance, was carried out online at Statistics Kingdom (https://www.statskingdom.com/pca-calculator.html; accessed on 1 December 2024). The Jaccard index was computed as reported earlier [28].

## 3. Results

### 3.1. Optical Microscopy of Soybean Components

Imaging the distribution of chemical constituents in soybeans by optical microscopy requires prior knowledge of the spectra of pure soybean constituents. Different biochemical substances can be visualized differently under microscopy according to the autofluorescence. Our results showed that the transverse sections of soybean seeds were highly fluorescent under the laser confocal microscope, indicating that several substances with autofluorescence were present in the observed varieties (Figure 1, Figure 2 and Figure 3). We propose that the blue fluorescence in the observed soybean varieties’ seeds was due to the presence of phenolic compounds such as hydroxycinnamic acids [29]. Within this class of compounds, ferulic acid was the major contributor to the blue fluorescence; however, other compounds in this class, such as p-coumaric and caffeic acids, have also been associated with such fluorescence [30]. Other than hydroxycinnamic acids, lignin has also been associated with blue fluorescence in plant tissues [31]; however, the observed blue fluorescence is mostly due to hydroxycinnamic acids. This is because of the fact that legume seed coats generally contain low lignin contents [32,33]. However, this is not strictly associated with seed coats as their cotyledons are also poorly lignified [34]. On the other hand, the soybean seeds were rich in secondary metabolites such as flavonoids, alkaloids, phenols, and others. However, flavonols were characterized by green rather than blue autofluorescence [35,36], as noted in our results when the soybean samples were excited with 500 to 545 nm. Finally, we also noted the red fluorescence (Figure 2), which was associated with anthocyanins and anthocyanidins [37,38]. These results were further confirmed by MS, as presented in the next sections (Table 1 and Table 2). Fluorescent flavonoids or their oxidation products, e.g., *Lignum nephriticum* (matlaline), have long been reported [39]. However, they also emit yellow and orange autofluorescence. Considering their importance in the health industry as well as for plants’ resistance to biotic and abiotic stresses, our results are important for their detection. Particularly in plants, they play an important role in auxin transport, root and shoot development, the control of reactive oxygen species, pollination, symbiotic nodule formation, and acting as protective compounds [40]. Together with hydroxycinnamic acids and other weakly autofluorescent phenolic substances, flavonoids are responsible for the fluorescence of the leaf epidermis [41,42]. Therefore, consistent with our earlier work on three *Glycine* species [27] and other reports on, e.g., paprika [36] and Arabidopsis [43,44], the utility of optical microscopy in the detection and distribution of these compounds in different plant tissues is valuable.

Figure 2A illustrates a multispectral image of the transverse section of the soybean variety “Locus” (Russia), displayed across all measured spectra. Figure 2B illustrates a spectral image in a blue color that indicates the presence of hydroxycinnamic acids in the soybean variety “Locus” (Russia). The spectral image in the red color indicates the presence of anthocyanin content in the soybean variety “Locus” (Russia) (Figure 2C). The microscopic analysis showed that the seed coat of this black-seeded variety had the brightest red fluorescence. Interestingly, it has been previously reported that the black color of the seed coat in legumes is due to the accumulation of anthocyanins [45]. This confirms that bright red fluorescence is caused by such compounds.

As shown in Figure 3B, we observed a much more pronounced presence of hydroxycinnamic acids in the “Namul” soybean variety than in the “Musson” soybean variety. A lighter blue fluorescence with more pronounced green, as shown in Figure 3C, indicated relatively lower hydroxycinnamic acids and lignin in “Primorskaya-4” compared with “Namul” and “Primorskaya 86” (Figure 3C). Thus, the autofluorescence enabled us to predict and determine their distribution across different sections of the observed tissue.

Figure 4A demonstrates that the soybean variety “Sphere” was much richer in anthocyanin content than the soybean varieties “Primorskaya 4” and “Primorskaya-86”. Figure 4B represents a spectral image in a green color that indicates the presence of flavonols in the soybean variety “Sphere”. Figure 4C represents a spectral image in a red color that indicates the presence of anthocyanin content in the soybean variety “Sphere”.

### 3.2. Tandem Mass Spectrometry Analysis

We further analyzed the soybean seed extracts by tandem mass spectrometry to better capture the diversity of phytochemicals. Our results revealed that all the studied soybean varieties were rich in bioactive compounds; sixty-nine polyphenolic compounds were tentatively identified and characterized by comparing fragmentation patterns and retention times. The chemical constituents were identified by comparing their retention indices, mass spectra, and MS fragmentation with an in-house self-built database (Biotechnology, Bioengineering and Food Systems Laboratory, Far-Eastern Federal University, Russia). The in-house self-built database was based on data from other spectroscopic techniques, such as nuclear magnetic resonance, ultraviolet spectroscopy, and MS, as well as data from the literature, which is continuously updated and revised. The capture rate was one spectrum for MS and two spectra for MS/MS. Data acquisition was controlled by Windows software for BRUKER DALTONIKS. All experiments were repeated three times. A four-stage ion separation mode (MS/MS mode) was implemented.

All the tentatively identified compounds along with molecular formulas, *m*/*z* calculated and observed, MS/MS data, and their comparative profiles for soybeans (eight varieties) are summarized in Appendix A, Table A1. Overall, one hundred and fourteen compounds belonging to different compound classes were detected from the eight soybean varieties. There were no commonly detected metabolites between the eight soybean varieties (Figure 5A). Principal component analysis indicated that the “Namul” and “Musson” varieties were quite similar, whereas “Primorskaya-4”, “Sphere”, and “Breeze” were grouped closer to each other. Similarly, “Primorskaya-86”, “Primorskaya-96”, and “Locus” were grouped closer to each other (Figure 5B). Several tentatively identified CID spectra (collision-induced spectrum) of chemical compounds in the soybean varieties “Primorskaya-4”, “Primorskaya-86”, “Primorskaya-96”, “Locus”, “Sphere”, “Breeze”, “Namul”, and “Musson” are presented below (Figure 5C–E). 

The highest number of tentatively identified compounds from the polifenolic class were classified as flavones (32), followed by phenolic acids (10), flavonols (9), flavan-3-ols (5), anthocyanidins (4), lignans (4), condensed tannins (2), coumarins (2), dihydrochalcone, and a stilbene. Moreover, fifty chemical compounds of other classes were identified, some of which were identified for the first time, e.g., 9,10-dihydroxy-8-oxooctadec-12-enoic acid and 13-trihydroxy-octadecenoic acid and the compound sterol class desmosterol. Among the studied soybean varieties, “Locus” contained the richest polyphenolic content. Next, “Primorskaya-86” was the second richest in polyphenols, with twenty-eight compounds (Figure 5A). Under the same experimental conditions, we could identify only nine polyphenolic compounds from the “Sphere” variety. Among the identified polyphenolic compounds in the studied soybean varieties, seventeen (flavones, flavonols, anthocyanins, phenolic acids, etc.) were commonly found in all soybean varieties. Principal component analysis indicated that there was 22.61% and 18.18% variability, as displayed by principal components 1 and 2, respectively (Figure 5B). Generally, we observed that “Namul” and “Musson” were grouped together. “Primorskaya-4”, “Sphere”, and “Breeze” were grouped together. The varieties “Primorskaya-96”, “Locus”, and “Primorskaya-86” were grouped together, indicating that they possibly have similar polyphenol compositions.

Figure 5C–E show examples of the decoding spectra (collision-induced dissociation (CID) spectrum) of the ion chromatogram obtained using tandem MS. The mass spectrum in negative ion mode of kaempferol from extracts of the soybean variety “Primorskaya 4” is shown in Figure 5C. The [M − H]^−^ ion produced three fragment ions at *m*/*z* 257.27, *m*/*z* 185.21, and *m*/*z* 117.27 (Figure 5C). The fragment ion with *m*/*z* 185.21 yielded one daughter ion at *m*/*z* 117.26. Mass spectrometry of kaempferol is presented in detail in scientific studies on *Juglans mandshurica* [46], *Polygala sibirica* [47], *Rhus coriaria* [48], *Lonicera japonica* [49], *Ribes meyeri* [50], andean blueberry [51], potato [52], and potato leaves [53].

The mass spectrum in positive ion mode of daidzein from extracts of the soybean variety “Locus” is shown in Figure 5D. The [M + H]^+^ ion produced two fragment ions at *m*/*z* 199.15 and *m*/*z* 137.12 (Figure 5D). The fragment ion with *m*/*z* 199.15 yielded two daughter ions at *m*/*z* 181.16, and *m*/*z* 129.24. Mass spectrometry of daidzein is presented in detail in scientific studies on black soya [54], soybean [55], *Hedyotis diffusa* [56], and *Loropetalum chinense* [57].

The mass spectrum in positive ion mode of daidzin from extracts of the soybean variety “Primorskaya-86” is shown in Figure 5E. The [M + H]^+^ ion produced one fragment ion at *m*/*z* 255.15 (Figure 5E). The fragment ion with *m*/*z* 255.15 yielded three daughter ions at *m*/*z* 199.18, *m*/*z* 227.20, and *m*/*z* 137.14. This bioactive substance was identified in mass spectrometric studies of extracts of black soya [54] and *Malus toringoides* [58].

Also, to present the similarities and differences in bioactive substances in different varieties of soybeans, we used the Jaccard index. Table 1 below presents the Jaccard index calculated for the sum of chemical compounds present in the soybean varieties.

Table 2 below presents the occurrence of identified chemical substances in the studied soybean varieties (“Primorskaya-4”, “Primorskaya-86”, “Primorskaya-96”, “Locus”, “Sphere”, “Breeze”, “Namul”, “Musson”).

## 4. Discussion

Soybean is an important legume and oil crop in the world because of the provision of food, feed, and industrial products and co-products. Soybean seeds are rich in secondary metabolites such as flavonoids, isoflavones, saponins, amino acids and their derivatives, anthocyanins, phenolic acids, hydroxycinnamic acids, lignans and coumarins, nucleic acids and their derivatives, carbohydrates, and lipids [59,60,61]. Due to the presence of such a diverse range of secondary metabolites, research on their beneficial health effects has significantly improved our understanding of their utility as functional foods [62]. With the increasing number of soybean varieties suitable for the specific agricultural zones of Russia, it is important to understand the secondary metabolite composition of the varieties so that consumers are well informed. The detection of these compounds by traditional biochemical techniques, such as HPLC and MS, should also be supplemented with affordable and reliable techniques that can provide quick and basic knowledge about the possible secondary metabolite composition of soybean seeds. In this regard, our results obtained in this study highly suggest that optical microscopy followed by the use of HPLC and MS are useful for both local soybean breeders, farmers, industrialists, and consumers.

Molecules in living tissues of plant matrices will produce fluorescent radiation when excited by appropriate wavelengths. This has been successfully used in the development of spectral markers to understand the metabolic composition of important crops [63]. Some researchers use this technique to understand whether plant tissues are under biotic stress by imaging the increase or decrease of specific metabolites, such as lignin, during infection by *Pythium ultimum* in apple roots [64]. Our results in soybean are quite useful for the development of fluorescent markers to understand the metabolomic composition of seeds. In particular, the results that we detected hydroxycinnamic acids and lignin (blue fluorescence), flavonols and their derivatives (green fluorescence), anthocyanins, anthocyanidins (red fluorescence), kaempferol, and quercetin (Figure 1, Figure 2 and Figure 3; Table 1). These results are based on previous findings that phenolic hydroxycinnamic acids such as hydroxycinnamic acids (e.g., p-coumaric and caffeic acids) are responsible for the fluorescence [29,30]. Similarly, the fact that lignin produces blue fluorescence in plant tissues [31] suggests that the fluorescence microscopy of soybean seeds can be used as a kind of marker to quickly estimate the possible lignin content in seeds. The presence of different lignin contents can also be revealed by such markers, even if the seed lignin content is low [32,33,34]. Soybeans are mainly consumed because of their high flavonoid content, including isoflavonoids, flavonols, flavanols, and others. Therefore, our microscopic results can become a useful tool in soybean-related industries to detect the estimates of these secondary metabolites [35,36]. In addition to these major metabolites in soy, some soybean varieties are rich in pigment compounds such as anthocyanins or anthocyanidins, which are also associated with a number of health benefits. The red spectrum due to these metabolites may therefore be further developed as a useful detection strategy [37,38].

Nevertheless, for the detailed metabolomic composition of secondary metabolites in soybean seeds, techniques such as HPLC and tandem MS are still preferred.

Many publications have shown that the structures of polyphenolic compounds correlate with their bioactivity [65,66]. Therefore, it is of utmost importance to correctly identify the molecular structures using a time-saving method such as mass spectrometry. Electrospray ionization tandem mass spectrometry (ESI-MS/MS) has been widely used for the structural characterization of both flavonoids and other chemical compounds, including the characterization of the aglycone structure and glycan sequence [67,68,69]. Flavonoids have been extensively analyzed for isomeric differentiation and structural characterization using ESI-MS/MS methods in positive and/or negative ion modes [70,71,72]. Electrospray ionization (ESI) coupled with tandem mass spectrometry (MS/MS) has proven to be a valuable method with high sensitivity and high resolution. So far, there have been many reports on the fragmentation characteristics of flavonoid group isomers using tandem mass spectrometry [73]. Rich structural information can be obtained by collision induced dissociation (CID) during MS/MS analysis. The elemental composition of both precursor and product ions can be obtained from high-resolution mass spectrometry (HRMS/MS) analysis. Thus, the product ion structures obtained from MS/MS experiments can be confirmed.

Our results that the studied Russian soybean varieties contained one hundred and fourteen different secondary metabolites indicate their richness and potential use as functional foods. Although the number of detected compounds in the studied varieties was lower than that reported for several Chinese soybean varieties [74], our results are still valuable for local soybean breeders to further improve these varieties by manipulating specific genes/pathways. Due to differences in their genetic background, the environment in which they are grown, and how the seeds are handled after harvest, the observed variation in the soybean varieties studied could have been due to any of these factors. Also, since we grew these soybean varieties under similar agronomic conditions, it is understandable that the observed differences are mainly due to their genetic background. Studies by other researchers using multiple soybean cultivars have shown that the metabolomic composition differs between cultivars. For example, a study using 29 Chinese soybean varieties detected 169 metabolites and reported that 104 of them showed intervarietal variation [75]. The metabotyping of different Korean soybean varieties showed that the metabolomic profiles of cultivated, wild, and semi-wild soybeans differed. Even within cultivated soybeans, the metabolomic composition differed. The authors related these differences to the distinct adaptation of the studied plant material to its respective environmental conditions [76]. Overall, we conclude that the studied cultivars differ in their metabolomic composition due to their genetic background and can therefore be used differently due to the composition and presence of certain health-promoting compounds.

## 5. Conclusions

The present study addressed the important question of the relative contribution of genotype to light anthocyanin color in soybeans. Such important traits as soybean color and anthocyanin content are tightly controlled by genotype, allowing for a wide range of selection. Our results based on metabotyping and fluorescence microscopy highlight that the “Locus” variety is good in terms of polyphenolic composition, followed by the variety “Primorskaya-86”. Our results are highly relevant for the development of fluorescent markers for the early detection of secondary metabolites. In addition, our results on the detection of health-promoting compounds (secondary metabolites) from these eight soybean varieties are highly relevant to the ongoing soybean breeding programs aimed at the development of nutritionally rich varieties.

## Figures and Tables

**Figure 1 plants-14-01995-f001:**
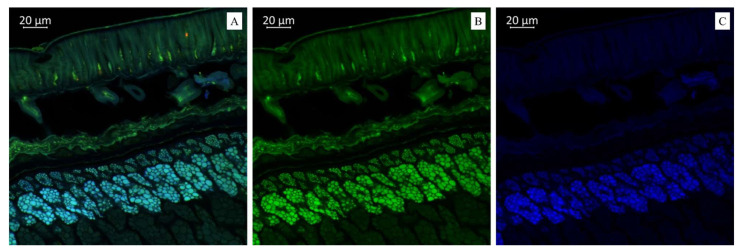
(**A**). Multispectral image of a cross section of soybean variety “Breeze” (Russia), presented in all measured spectra. Excitation at 405 nm with emission in the range of 400–475 nm (blue); excitation at 488 nm with emission in the range of 500–545 nm (green) and 620–700 nm (red). (**B**). Presence of flavonols (green color). (**C**). Presence of hydroxycinnamic acids (blue color).

**Figure 2 plants-14-01995-f002:**
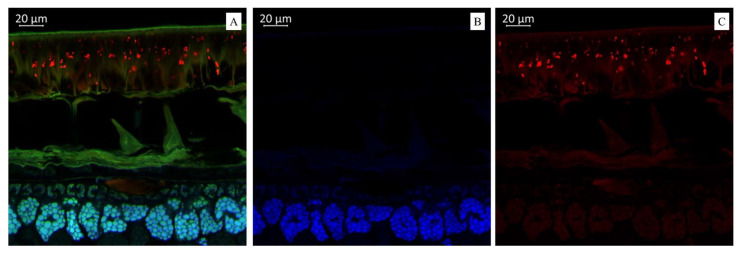
(**A**). Multispectral image of transverse section of the soybean variety “Locus” (Russia), presented in all measured spectra. Excitation at 405 nm with the emission in the range of 400–475 nm (blue); excitation at 488 nm with the emission in the range of 500–545 nm (green) and 620–700 nm (red). (**B**). Presence of hydroxycinnamic acids (blue color). (**C**). Presence of anthocyanin content (red color).

**Figure 3 plants-14-01995-f003:**
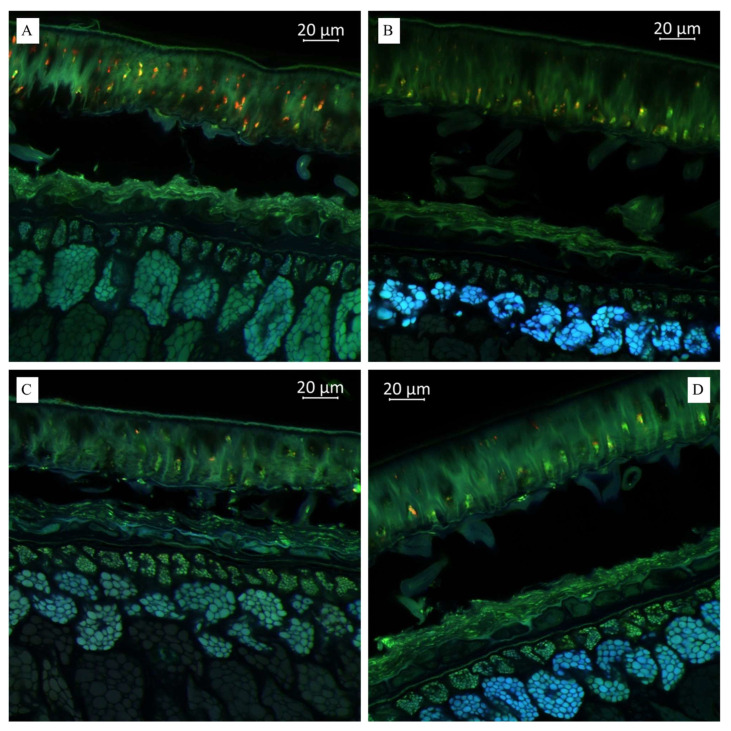
(**A**). Multispectral image of transverse section of the soybean variety “Musson” (Russia), presented in all measured spectra. Excitation at 405 nm with the emission in the range of 400–475 nm (blue); excitation at 488 nm with the emission in the range of 500–545 nm (green) and 620–700 nm (red). (**B**). Multispectral image of transverse section of the soybean variety “Namul” (Russia), presented in all measured spectra. (**C**). Multispectral image of transverse section of the soybean variety “Primorskaya 4” (Russia). (**D**). Multispectral image of transverse section of the soybean variety “Primorskaya 86” (Russia).

**Figure 4 plants-14-01995-f004:**
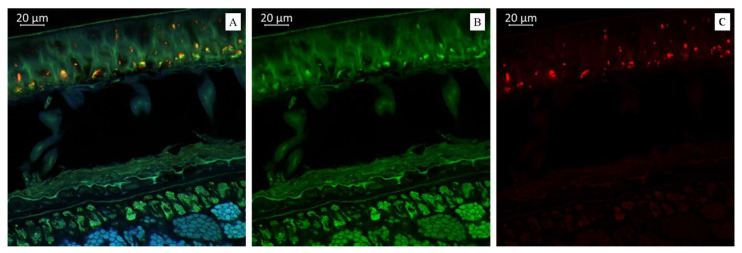
(**A**). Multispectral image of transverse section of the soybean variety “Sphere” (Russia), presented in all measured spectra. Excitation at 405 nm with the emission in the range of 400–475 nm (blue); excitation at 488 nm with the emission in the range of 500–545 nm (green) and 620–700 nm (red). (**B**). Presence of flavonols. (**C**). Presence of anthocyanin content in the soybean variety “Sphere” (Russia).

**Figure 5 plants-14-01995-f005:**
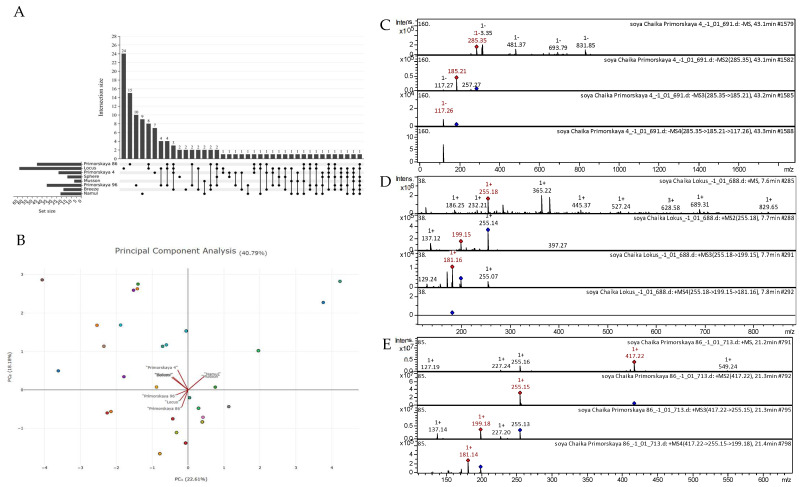
(**A**) Upset plot showing similarities and differences in the presence of polyphenol group in soybean varieties. (**B**) Principal component analysis based on the presence/absence of the detected metabolites in studied soybean varieties. (**C**) CID spectrum of kaempferol from extracts of soybean variety “Primorskaya-4”, *m*/*z* 285.35. At the top is an MS scan in the range of 100–1700 *m*/*z*; at the bottom are fragmentation spectra (from top to bottom): MS2 of the protonated kaempferol ion (285.35 *m*/*z*, red diamond), MS3 of the fragment 285.35→185.21 *m*/*z*, and MS4 of the fragment 285.35→185.21 →117.26 *m*/*z*. (**D**) CID spectrum of daidzein from extracts of soybean variety “Locus”, *m*/*z* 255.18. At the top is an MS scan in the range of 100–1700 *m*/*z*; at the bottom are fragmentation spectra (from top to bottom): MS2 of the protonated daidzein ion (255.18 *m*/*z*, red diamond), MS3 of the fragment 255.18→199.15 *m*/*z*, and MS4 of the fragment 255.18→199.15 →181.16 *m*/*z*. (**E**) CID spectrum of daidzin from extracts of soybean variety “Primorskaya-86”, *m*/*z* 417.22. At the top is an MS scan in the range of 100–1700 *m*/*z*; at the bottom are fragmentation spectra (from top to bottom): MS2 of the protonated daidzin ion (417.22 *m*/*z*, red diamond), MS3 of the fragment 417.22→255.15 *m*/*z*, and MS4 of the fragment 417.22→255.15 →199.18 *m*/*z*.

**Table 1 plants-14-01995-t001:** Jaccard index for eight soybean varieties of the polyphenol group (“Primorskaya-4”, “Primorskaya-86”, “Primorskaya-96”, “Locus”, “Sphere”, “Breeze”, “Namul”, “Musson”).

	Locus (66)	Namul (21)	Musson (41)	Sphere (16)	Breeze (20)	Primorskaya-86 (43)	Primorskaya-4 (22)	Primorskaya-96 (43)
**Locus** (66)		7 0.0875	20 0.2299	11 0.1549	14 0.1944	24 0.2824	11 0.1429	24 0.2824
**Namul** (21)	7 0.0875		12 0.2400	5 0.1563	3 0.0789	3 0.0492	6 0.1622	6 0.1034
**Musson** (41)	20 0.2299	12 0.2400		6 0.1176	8 0.1509	16 0.2353	9 0.1667	18 0.2727
**Sphere** (16)	11 0.1549	5 0.1563	6 0.1176		7 0.2414	5 0.0926	8 0.2667	10 0.2041
**Breeze** (20)	14 0.1944	3 0.0789	8 0.1509	7 0.2414		9 0.1667	8 0.2353	10 0.1887
**Primorskaya-86** (43)	24 0.2824	3 0.0492	16 0.2353	5 0.0926	9 0.1667		5 0.0833	18 0.2647
**Primorskaya-4** (22)	11 0.1429	6 0.1622	9 0.1667	8 0.2667	8 0.2353	5 0.0833		11 0.2037
**Primorskaya-96** (43)	24 0.2824	6 0.1034	18 0.2727	10 0.2041	10 0.1887	18 0.2647	11 0.2037	

**Table 2 plants-14-01995-t002:** The occurrence of identified chemical substances in the studied soybean varieties (“Primorskaya-4”, “Primorskaya-86”, “Primorskaya-96”, “Locus”, “Sphere”, “Breeze”, “Namul”, “Musson”).

Chemical Substances	Occ.	Present in Soybean Varieties
Myristoleic acid	7	Locus, Namul, Musson, Sphere, Breeze, Primorskaya-86, Primorskaya-96
Acacetin	6	Locus, Namul, Musson, Breeze, Primorskaya-86, Primorskaya-96
Daidzin	6	Locus, Musson, Sphere, Breeze, Primorskaya-86, Primorskaya-96
Sucrose	6	Locus, Musson, Breeze, Primorskaya-86, Primorskaya-4, Primorskaya-96
Trehalose	6	Locus, Musson, Breeze, Primorskaya-86, Primorskaya-4, Primorskaya-96
Apigenin	5	Locus, Musson, Breeze, Primorskaya-86, Primorskaya-96
Caffeic acid derivative 1	5	Locus, Namul, Sphere, Primorskaya-4, Primorskaya-96
Ethyl protocatechuate	5	Musson, Breeze, Primorskaya-86, Primorskaya-4, Primorskaya-96
Genistein	5	Locus, Sphere, Breeze, Primorskaya-4, Primorskaya-96
L-Tryptophan	5	Locus, Musson, Sphere, Primorskaya-86, Primorskaya-96
Sespendole	5	Locus, Sphere, Breeze, Primorskaya-86, Primorskaya-4
Adenosine	4	Locus, Musson, Primorskaya-4, Primorskaya-96
Apigenin-7-O-glucoside	4	Locus, Sphere, Primorskaya-86, Primorskaya-96
Catechin	4	Locus, Musson, Primorskaya-86, Primorskaya-96
Daidzein	4	Locus, Musson, Primorskaya-86, Primorskaya-96
Kaempferol	4	Sphere, Breeze, Primorskaya-4, Primorskaya-96
Syringaresinol	4	Locus, Primorskaya-86, Primorskaya-4, Primorskaya-96
9,10-Dihydroxy-8-oxooctadec-12-enoic acid	3	Breeze, Primorskaya-86, Primorskaya-96
Acacetin O-glucoside	3	Locus, Musson, Primorskaya-96
Epiafzelechin	3	Locus, Sphere, Primorskaya-96
Formononetin	3	Locus, Namul, Musson
Genistein C-glucoside malonylated	3	Locus, Musson, Primorskaya-86
Glucoheptonic acid	3	Locus, Breeze, Primorskaya-4
Glycitein	3	Locus, Namul, Primorskaya-86
Glycitin	3	Locus, Musson, Primorskaya-96
Linolenic acid	3	Locus, Primorskaya-86, Primorskaya-96
Protocatechuic acid	3	Locus, Musson, Primorskaya-96
Punicalin alpha	3	Sphere, Primorskaya-4, Primorskaya-96
Rhamnetin-O-hexoside	3	Locus, Primorskaya-86, Primorskaya-96
Trehalose dihydrate	3	Locus, Musson, Primorskaya-96
(Epi)Gallocatechin	2	Locus, Primorskaya-86
11-Hydroperoxy-octadecatrienoic acid	2	Locus, Primorskaya-86
Acetyl daidzin	2	Primorskaya-86, Primorskaya-96
Acetyl genistin	2	Locus, Primorskaya-86
Cyanidin-3-O-glucoside	2	Locus, Primorskaya-86
Dihydroquercetin	2	Musson, Primorskaya-86
Dimethoxy-trihydroxy(iso)flavone	2	Namul, Musson
Ellagic acid pentoside	2	Locus, Primorskaya-96
Epiafzelechin derivative	2	Musson, Primorskaya-86
Herbacetin	2	Namul, Musson
Inosine	2	Locus, Primorskaya-4
Luteolin 7-O-glucoside	2	Locus, Primorskaya-86
Malic acid	2	Musson, Primorskaya-86
Malonyl daidzin	2	Primorskaya-86, Primorskaya-96
Methyl palmitoleate	2	Namul, Musson
Monopalmitin	2	Locus, Primorskaya-86
Pelargonidin-3-glucoside	2	Locus, Primorskaya-96
Phenylalanine	2	Musson, Primorskaya-96
Soyasaponin Bb′	2	Locus, Breeze
Soyasaponin I	2	Locus, Primorskaya-96
Trihydroxy(iso)flavone	2	Locus, Primorskaya-86
Tryptamine	2	Musson, Primorskaya-86
Vebonol	2	Locus, Breeze
Vitexin	2	Locus, Breeze

## Data Availability

All datasets produced as a result of this work are given in the main manuscript.

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
