# Peer review of "Autofluorescence and Metabotyping of Soybean Varieties Using Confocal Laser Microscopy and High-Resolution Mass Spectrometric Approaches"

_plants, 2025, doi:10.3390/plants14131995_

Round 1

Reviewer 1 Report

Comments and Suggestions for Authors

In my opinion the claimed identification of many compounds, in the manuscript plants-3494146, is unconvincing. I will describe only four examples.
1.    The identification of caffeic acid O-hexoside (page 8) is based on the product ions at m/z 179, 119 and 113. However this compound should yields product ion at m/z 179 and 135 (https://massbank.eu/MassBank/RecordDisplay?id=MSBNK-RIKEN-PR309375&dsn=RIKEN). Product ions at m/z 179, 119 and 113 are characteristic of sucrose (https://massbank.eu/MassBank/RecordDisplay?id=MSBNK-RIKEN_ReSpect-PT200600&dsn=RIKEN_ReSpec)
2.    For L-histidine the authors claim the product ion at m/z 147 (Appendix 1. Table 1.), which would corresponds to the loss of mass 9, which is nonsense.
3.    For soyasaponins, I would expect mainly the product ions formed through the breaking of glycosidic bonds, e.g. the loss of rhamnose moiety, i.e. the loss of mass 146 for soyasaponin Bb (10.1039/D2FO00537A).
4.    Compound No 122 (Appendix 1. Table 1.) has almost identical calculated mass and observed mass of [M+H]+.

Author Response

Dear Reviewer.

Thank You very much for the time spent on reviewing our scientific work.

We, together with the team of authors, have reviewed in detail your indications of errors and corrections, and are very grateful to you for the inaccuracies indicated.

  1. Regarding hexoside-O-caffeic acid, yes, we have indeed reviewed all the mass spectra for all six samples in which a similar spectrum is found (Locus, Musson, Breeze, Primorskaya-86, Primorskaya-4, Primorskaya-96). And we have indeed attributed it to sucrose. There is no product ion m/z 135 anywhere. This is our oversight. Thank you for the error indicated.
  2. This is a technical error. We have excluded this item from the table.
  3. Regarding the identification of soyasaponin Bb’, our opinions are divided.

Wu et al (2008) in their article “Comparative Metabolic Profiling Reveals Secondary Metabolites Correlated with Soybean Salt Tolerance” provide the following decoding of the ion products of soyasaponin Bb’: gal (beta-D-galactopyranosyl)(1-2)glc UA(beta-D-glucoronopyranosyl) (1-3) : 751, 633, 615, 457

Hu et al (2017) in their article “Differences in the metabolic profiles and antioxidant activities of wild and cultivated black soybeans evaluated by correlation analysis” also identify soyasaponins and, again referring to Wu et al (2008), provide fragmented ions 751, 633, 457.

In our studies, we obtained the following spectra: (The spectra are attached below in a Word file).

These mass spectra, you see, began to be identified after 30 minutes of the experiment, when, in our experience, all saponins and chikusetsusaponins begin to appear.

We identify these repeating mass spectra as soyasaponin Bb’.

We do not insist, it is very important for us to find the truth in this matter.

  1. This is an error. Everything has been corrected.

Reviewer 2 Report

Comments and Suggestions for Authors

Recommend the author state ‘tentative identification” for MS/MS experiments. The experiments did not quantify the compounds, the authors should complement these data or state that this is a limitation of the work.

Author Response

Dear Reviewer.
Thank You very much for your comments and for your time.

Our team of authors corrected everything in the text (corrections are highlighted in blue).
We also inserted an additional table into the text showing the identification of bioactive substances in different varieties of soybeans studied.

Reviewer 3 Report

Comments and Suggestions for Authors

Overall, this is an excellent manuscript which demonstrates utilizing inherent autofluorescence of compounds inherent in soybean seed cultivars to characterize them first with confocal laser microscopy, and then by liquid chromatography coupled to tandem mass spectrometry to provide a profile for each of eight different soybean cultivars grown in Russia. There exists considerable global interest in profiling different soybean cultivars according to their metabolite profiles, in particular to deduce whether molecular differences can account for phenotypic differences in the varieties. What is particularly keen about this research team is the combination of autofluorescence with tandem mass spectrometry. This is frankly some of the most exciting work in the field of soybean metabolomics, and this reviewer is very enthusiastic about this line of research and in particular this manuscript. I am particularly interested in the findings regarding anthocyanins as I believe we may have detected these ourselves in soybean leaves, but we have not done the MS/MS work needed to prove their structures as these authors have.

There are a few minor points the authors should address, yet this does not detract from a well-written and conceived manuscript. I would be delighted to review additional papers in the field from this team--their work is very likely helpful to our own research program (hence why I am signing the review).

Minor points:

  1. I think it would be preferable to avoid using "high-resolution" in the title, because most in the field of mass spectrometry would not consider a Bruker ion trap instrument to be particularly high-resolution. Instead, I would emphasize that they use "Tandem" mass spectrometry, because this is the feature that allows for identification of the molecular structures. My group does use high-resolution to identify soybean leaf metabolites, but alone this only good for identification of molecular formulas--it is the tandem mass spectrometry that validates the assignments, and I feel this is the aspect the authors should emphasize.
  2. In the Abstract Figure legend, line 35, use "mass spectrometry" as opposed to "mass spectroscopy" because spectroscopy implies using photons.
  3. Line 166. Ref 18 is duplicated here.
  4. Line 216.  Should be "HPLC-MS/MS"
  5. Line 293. This sentence is awkward and hard to understand. Perhaps replace with "When excited with appropriate wavelengths, molecules within living tissues will produce fluorescence emission."

Author Response

Dear Reviewer.

First of all, our team of authors expresses their utmost appreciation for the revision of our work and the time You spent on us.

1. Yes, we completely agree with your correct criticism and will try to negotiate with the editors in a separate letter to slightly change the title, if the editors allow us and this does not entail a problem with resubmitting the article.
2. This inaccuracy has been corrected.
3. This omission has also been corrected.
4. The sentence has been formatted and highlighted in blue. There are also quite a few corrections in the article, highlighted in blue, and the article has been supplemented with a table showing the presence of bioactive substances by the varieties of soybean samples studied.

Reviewer 4 Report

Comments and Suggestions for Authors

The manuscript entitled: “Autofluorescence and Metabotyping of Soybean Varieties Using Confocal Laser Microscopy and High-Resolution Mass Spectrometric Approaches” could be published in Plants, but after a major revision.

This study investigated a detailed metabolomic and comparative analysis of bioactive substances of selected soybean varieties. Tandem mass spectrometry was used to identify chemical constituents in soybean extracts. The results of initial studies revealed the presence of one hundred and twenty-eight compounds, seventy-seven of the target analytes were tentatively identified as compounds from polyphenol group.

However, the manuscript needs some important clarifications and revision.

  1. Introduction needs to be extended, including previous LC-MS profiling of soybean and the benefits of the used methods.
  2. The main disadvantage of the work is the unreliable identification of the metabolites. The authors must check carrefylly all positive and negative ion modes (For example: rows 257-260” The mass spectrum in positive ion mode of aromadendrin 7-O-rhamnosid” …Than it is written [M - H])-
  3. Using LC-MS without comparison with reference standards, the annotation of “rhamnoside, glucoside etc” is not possible. Use deoxyhexose, hexose, etc.
  4. It is not clear how the authors have distinguished Jaceosidin, from Cirsiliol and Dimethoxy-trihydroxy(iso)flavone; kaempferol from luteolin; chlorogenic from neochlorogenic acid. In my opinion, the neo is 4-caffeoylquinic acid. There is no reliable discussion concerning MS/MS fragmentation. With regards of saponins is the same. The whole LC-MS analysis needs an in-depth revision.
  5. 5. High-Resolution Mass Spectrometry needs the masses and fragments to be written at least to the fourth decimal place

Author Response

Dear Reviewer.

The team of authors expresses deep gratitude to you for your work on the article and your time spent.
1. We carefully rechecked all the results and removed a fairly large number of controversially (in your opinion) identified chemical components. In our identifications of chemical compounds, we relied on both our developed database and mass spectrometric articles by many authors (all of them are indicated in the links), we tried to find articles devoted to soy or legumes.
2. Regarding aromadendrin (this was a technical error), jaseosidin and neochlorogenic acid: we also removed these controversial identifications from the table.
3. We also significantly reworked the entire text of the article, removed some things, highlighted new inserts in blue, significantly added articles by authors on mass spectrometry, to which we refer in the text of the article.

Round 2

Reviewer 1 Report

Comments and Suggestions for Authors

In my opinion the manuscript has not been improved to the acceptable level. The claimed identification of many compounds is unconvincing. I will describe only three examples (but there are many more in the manuscript).
1.    Figure 5C, loss of mass 120 is a characteristic feature of flavonoid C-glycoside (e.g. 10.1016/j.ijms.2012.08.035, 10.1177/1469066720963003, 10.1002/jms.3413), not flavonoid O-glycosides.
2.    For L-Lysine the authors claim the product ion at m/z 137 (compound No 83, Appendix 1. Table 1.), which would corresponds to the loss of mass 10, which is nonsense.
3.    Compound No 123 (Appendix 1. Table 1.) - soyasaponin Bb' (compound mentioned in the authors’ response!), has calculated mass 796.981, which is nonsense.

Author Response

Dear Reviewer.

The team of authors expresses their deep gratitude to you for your work on the article and your time spent.

1. We carefully rechecked all the results and removed a fairly large number of controversially identified chemical components. We also studied and took into account three articles mentioned by you, and we refer to them in our text. In our identifications of chemical compounds, we relied on both our own developed database and mass spectrometric articles by many authors (all of them are indicated in the references), we tried to find articles devoted to soy or legumes.
2. Regarding L-lysine and other controversial compounds: we also removed these controversial identifications from the table.
3. We also significantly reworked the entire text of the article, removed some things, highlighted new inserts in blue, significantly added articles by authors on mass spectrometry, to which we refer.
4. Regarding the mass of soy saponin Bb’, we have clarified it, here is the calculated mass:
https://www.invivochem.com/soyasaponin-iii.html

Reviewer 2 Report

Comments and Suggestions for Authors

I suggest to approve the manuscript after the authors` corrections.

Author Response

Dear Reviewer.
The authors' team would like to express their deep gratitude to you for your work on the article and your time spent.
1. We carefully rechecked all the results and removed a number of controversially identified chemical compounds. In our identifications of chemical compounds, we relied on both our own developed database and mass spectrometric articles by many authors (all of them are listed in the references), we tried to find articles devoted to soy or legumes.
2. We also significantly reworked the entire text of the article, removed some things, highlighted new inserts in blue, and significantly added articles by authors on mass spectrometry, to which we refer.

Reviewer 4 Report

Comments and Suggestions for Authors

Dear Authors,

The manuscript entitled: “Autofluorescence and Metabotyping of Soybean Varieties Using Confocal Laser Microscopy and High-Resolution Mass Spectrometric Approaches” could be published in Plants, after major revision.

The authors have revised their manuscript according to some of my previous suggestions, but some points need clarifications:

1.The authors must rearrange Materials and methods, and results and Discussion according to the Plants template requirements.

  1. m/z should be in italic.
  2. Appendix 1. Table 1. The masses and fragments should to be written at least to the fourth decimal place
  3. Again the identification of some metabolites is unreliable. Using LC-MS it is impossible to distinguish different sugars (glucoside, rhamnoside, etc). Without reference standards, it is better to use apigenin-O-hexoside than apigenin-O-glucoside. Please, revise the Table 1 appendix 1.

Author Response

Dear Reviewer.

Many thanks for your work on the article, your valuable time spent and pointing out possible errors in the article.
The team of authors almost completely rewrote the text of the article regarding mass spectrometric studies. Additional explanatory text in the article is highlighted in blue.
We also very critically reviewed the table of identified compounds and removed compounds from it that, upon closer examination, caused us possible doubts.

We very positively perceive your criticism and are always ready to cooperate.
Thank you very much again.

Round 3

Reviewer 4 Report

Comments and Suggestions for Authors

The manuscript was corrected according to the reviewer`s comments.